# 7p22.2 Microduplication: A Pathogenic CNV?

**DOI:** 10.3390/genes14061292

**Published:** 2023-06-19

**Authors:** Alessia Bauleo, Alberto Montesanto, Vincenza Pace, Francesco Guarasci, Rosalbina Apa, Rossella Brando, Laura De Stefano, Simona Sestito, Daniela Concolino, Elena Falcone

**Affiliations:** 1BIOGENET, Medical and Forensic Genetics Laboratory, 87100 Cosenza, Italy; bauleo@biogenet.it (A.B.); vincenza_pace@yahoo.it (V.P.); guarasci.francesco@gmail.com (F.G.); rosa.apa@hotmail.it (R.A.); rossellabrando@yahoo.it (R.B.); lauradestefano@virgilio.it (L.D.S.); falcone@biogenet.it (E.F.); 2Department of Biology, Ecology and Earth Sciences, University of Calabria, 87036 Rende, Italy; 3Pediatric Unit, Department of Science of Health, Magna Graecia University of Catanzaro, 88100 Catanzaro, Italy; sestitosimona@unicz.it (S.S.); dconcolino@unicz.it (D.C.)

**Keywords:** 7p22.2 microduplication, minimal critical region, neurodevelopmental phenotypes, CNV, array-CGH. 7p22.1, 7p22, SDK11

## Abstract

Partial duplication of the short arm of chromosome 7 is a rare chromosome rearrangement. The phenotype spectrum associated with this rearrangement is extremely variable even if in the last decade the use of high-resolution microarray technology for the investigation of patients carrying this rearrangement allowed for the identification of the 7p22.1 sub-band causative of this phenotype and to recognize the corresponding 7p22.1 microduplication syndrome. We report two unrelated patients that carry a microduplication involving the 7.22.2 sub-band. Unlike 7p22.1 microduplication carriers, both patients only show a neurodevelopmental disorder without malformations. We better characterized the clinical pictures of these two patients providing insight into the clinical phenotype associated with the microduplication of the 7p22.2 sub-band and support for a possible role of this sub-band in the 7p22 microduplication syndrome.

## 1. Introduction

Partial duplication of the short arm of chromosome 7 is a rare chromosome rearrangement associated with a clinical picture including high-frequency hypotonia, intellectual disability, autism, cardiovascular and skeletal abnormalities, and some craniofacial dysmorphisms [1,2,3,4,5,6]. The characterization of 7p microduplication in patients carrying this rearrangement allowed researchers to identify the 7p22.1 sub-band causative of this phenotype and to recognize the corresponding 7p22.1 microduplication syndrome [7,8,9]. Until today, the 7p22.1 seemed to be the only sub-band in the 7p22 region clearly associated with a syndromic phenotype. Very little data are currently available on duplications involving the 7p22.2 and 7p22.3 sub-bands and, consequently, their clinical significance is still unclear.

In 2015, Cox and Butler described a patient carrying a microduplication involving only the 7p22.2 sub-band showing a clinical phenotype only partially overlapping the 7p22.1 microduplication syndrome [10]. This patient showed a mild intellectual disability, asthma, myopia, proportionate short stature, dysmorphic features, and Achilles tendon release. Only a few cases of 7p22.3 rearrangement have been reported associated with a neurodevelopmental phenotype and heart anomalies, for which the genotype-phenotype correlation is still uncertain [11,12,13,14]. In particular, Mastromoro et al. described a four-year-old male child with a 139 kb deletion at 7p22.3 involving four different genes (FTSJ2, NUDT1, SNX8 and MAD1L1) that showed unilateral renal agenesis, learning and language delay, insomnia, attention deficit hyperactivity disorder, but not heart defects. Since these clinical characteristics were also reported for other 7p22 deletions cases involving ***SNX8***, the role of this gene in these neurodevelopment disorders was proposed.

Based on these lines of evidence, a better characterization of the distinct phenotypes associated with the 7p22 microduplication and a refinement of the involved genomic regions are required.

Here we report two new and unrelated patients that carry a microduplication exclusively involving the 7p22.2 sub-band. Both patients show a neurodevelopmental disorder and facial dysmorphisms but, unlike 7p22.1 microduplication carriers and similar to the reported case of Cox and Butler, do not exhibit congenital malformations. Given these patients constitute the second reported cases of a microduplication involving the 7.22.2 sub-band, we better characterized their clinical pictures providing insight into the clinical phenotype associated with the microduplication of the 7p22.2 sub-band.

## 2. Materials and Methods

### 2.1. Array-CGH

Genomic DNA was extracted from peripheral blood using a DNeasy Blood and Tissue kit (Qiagen, Inc., Valencia, CA, USA) on the QIAcube automated DNA extraction robot (Qiagen GmbH, Hilden, Germany) following the manufacturer’s protocol. Molecular karyotyping (array-CGH) was performed using the Agilent-California USA Human Genome CGH Microarray 4x180K platform according to the manufacturer’s protocol (Agilent Technologies, Santa Clara, CA, USA). Variant calling was performed using Agilent CytoGenomics Edition 5.0.2.5 (ADM-2 algorithm; release hg19).

All genomic positions were reported according to the human genome assembly (GRCh37/hg19) and subsequently confirmed using quantitative real-time PCR. The clinical interpretation of copy number variants (CNVs) was performed according to technical standard recommendations of the American College of Medical Genetics and Genomics (ACMG), using the semiquantitative system point-based scoring metric [15]. The classification was supported by use of different public databases, such as the Database of Genomic Variants (DGV), the Database of genomic variation and Phenotype in Humans using Ensemble Resources (DECIPHER), the Online Mendelian Inheritance in Man (OMIM), the University of California, Santa Cruz (UCSC) genome browser and an internal database based on our laboratory data.

### 2.2. Quantitative Real-Time PCR

Quantitative real-time PCR (qPCR) was performed on genomic DNA of patients and control samples with a negative chromosomal microarray profile in the tested genetic regions. Experiments were carried out in triplicates with a PowerUp SYBR Green Master mixture (Applied Biosystems, Foster City, CA, USA) in a QuantStudio 5 Real-Time PCR System (Applied Biosystems) and using a housekeeping gene (telomerase reverse transcriptase) as an internal reference. Target sequences for PCR experiments were retrieved from the USCS database. Primers used for each assay were designed using Primer Express 3.0 software and together with the genetic regions under investigation were reported in the Appendix A.

## 3. Results

### 3.1. Patient 1

A 4-year-old boy came to our attention after diagnosis of autism spectrum disorder (ASD). He was the second child of healthy non-consanguineous parents, born at term from uneventful pregnancy. His birth weight was 3.010 kg and no perinatal suffering was reported (APGAR score 8–9). During the first 2 years of life, his psychomotor development was normal, with the exception of an altered sleep-wake rhythm. A language regression at 2–2.6 years of age, in the social sphere at about 3 years of age, and lack of direct eye contact in the interactive games were subsequently reported. The cognitive development was within the normal range (Nonverbal IQ (NVIQ) = 93). Hyperactivity, limited eye contact, motor and vocal stereotypies, severe impairments in language and in communication, and fecal incontinence due to lack of sphincter control were also present. A facial dysmorphism without congenital malformations was also reported.

a-CGH analysis showed a microduplication located in the 7p22.2 region spanning 371.165 kb from position 3074785 to position 3445950. In addition, another microduplication spanning 345.669 kb from position 56670637 to position 57016306 was also detected at the 12q13.3 region. Real-time PCR experiments confirmed the presence of both CNVs. Both parents did not provide consent for further chromosomal analyses and, consequently, no segregation analysis was performed. Both CNVs have been classified as variants of uncertain significance (VUS) according to the ACMG recommendations.

### 3.2. Patient 2

A 4-year-old girl came to our attention for a psychomotor development delay with a significant impairment in the acquisition and use of language. She was born at term by caesarean section from non-sanguineous healthy parents after an uneventful pregnancy. Her birth weight was 2.2 kg and no perinatal suffering was reported (APGAR score 8–9). She was diagnosed with psychomotor development delay (trunk control at 8 months of age, standing at 18 months, autonomous walking at 30 months, lack of sphincter control at 36 months) with a significant impairment of the language area at 2.5 years of age. At the age of 3 years, she underwent inguinal hernia surgery. At 36 months a clinical evaluation revealed a neurodevelopmental disorder characterized by an expressive language impairment, motor coordination deficit; flat foot and knee valgus were noticed. Physical examination evidenced very mild dysmorphic traits including slightly upslanting palpebral fissures, relatively low-set protruding ears with prominent lobes, and hypoplastic alae nasi with depressed nasal tip. No congenital malformations were present. The audiological examination revealed no hearing problems although she was scared by loud noises. She was able to perform and imitate simple motor movements (i.e., jump, greet, use the fork) but showed some difficulties with the imitation of movement sequences. Moreover, she showed attention deficit not being able to stay focused for more than 15/20 min on a specific activity. Regarding sociality, she showed discomfort in socializing with peers, problems partially overcome when she started attending primary school.

a-CGH analysis showed a microduplication of 455.143 kb from position 2967525 to position 3422668 in the 7p22.2 region inherited from her unaffected father. Additionally, in this case, real-time PCR confirmed the presence of the detected CNV that was classified as a VUS according to the ACMG recommendations.

To help interpret the genotype-phenotype correlation in 7p22.2 duplications, the DECIPHER database (https://decipher.sanger.ac.uk/, accessing date: 2 November 2022) was queried and excluding patients who carry further CNVs, six subjects that had a copy number gain restricted to the only 7p22.2 sub-band were identified.

In Figure 1 and Table 1, the CNVs of these six patients together with the patient described in Cox and Butler (2015) and the two patients described in the present work were reported.

These patients carried microduplications ranging from 11 kb to 629 kb. Patient 292740 from DECIPHER shows a cognitive impairment and carries a microduplication comparable to that found in our patients and in the patient described in Cox and Butler. Three DECIPHER patients (292740, 266783 and 476695) have a microduplication involving only the SDK1 gene: two show only a neurodevelopmental phenotype, while for one of them (patient 476695) no phenotypic data were reported. The other three DECIPHER patients (274816, 401792 and 331820) carry a microduplication involving GNA12 and CARD11 genes but SDK1: these patients had a more complex and variable clinical phenotype but did not exhibit neurodevelopmental traits. All the microduplications involving SDK1 gene also include CpG islands located in the promoter or in the body gene. Patients carrying a microduplication involving SDK1 gene show a psychomotor and neurodevelopmental delay with a high frequency of some clinical traits such as speech delay, and dysmorphic traits. Cardiac abnormalities and major skeletal abnormalities were not reported, unlike what is usually reported in patients with 7p22.1 microduplication syndrome.

## 4. Discussion

The 7p22.1 microduplication syndrome is characterized by developmental delay, intellectual disability, and craniofacial dysmorphisms; cardiac and skeletal abnormalities are often present. The 7p22.1 sub-band has been proposed as the minimal region causing the “7p22 microduplication syndrome” [16]. Caselli et al., comparing the clinical phenotype of patients who underwent a-CGH investigation, identified the 7p22.1 sub-band as the critical region of the syndrome and better defined its clinical traits [7]. In the same year, Cox and Butler (2015), described a patient that carried a microduplication of the only 7p22.2 sub-band [10]. The duplicated region spanned 629 kb and included CARD11 and SDK1 genes and part of GNA12 gene. The phenotype of this clinical case only partially overlapped the clinical features of the 7p22.1 microduplication syndrome. He presented a history of developmental delay and mild intellectual disability, asthma, myopia, dysmorphic facial features, but did not show any cardiovascular abnormality and major congenital malformations. For this reason, Cox and Butler hypothesized that the 7p22.2 sub-band might contribute to the variability of the 7p22 phenotype. However, no other similar clinical cases have been reported in literature [10].

In this report we describe two children that carry a 7p22.2 microduplication overlapping the microduplication of the patient reported in Cox and Butler [10]. Both patients are affected by a marked speech delay, a motor deficit, and mild dysmorphic traits. In addition, an autism spectrum disorder (ASD) was recently diagnosed in the patient 1 and some minor skeletal abnormalities (flat feet and knee valgus) were found in patient 2. Patient 1 also carries a 12q13.3 microduplication but none of the genes contained in this region resulted to be disease-causing when present in an extra copy (triple-sensitivity) and no further literature data on this microduplication were available. For this reason, this microduplication currently does not appear to be of clinical relevance. In fact, patients here described and the patient of Cox and Butler show a milder clinical phenotype than 7p22.1 microduplication carriers. They had a neurodevelopmental disorder, facial dysmorphisms and minor skeletal abnormalities, but, unlike 7p22.1 microduplication carriers, do not exhibit major congenital malformations.

Currently, little is known about the 7p22.2 rearrangements and their correlation with pathological manifestations. The 7p22.2 sub-band is 1.7 Mb long and includes the caspase recruitment domain family, CARD11 gene (*607210), the sidekick cell adhesion molecule 1, SDK1 gene (* 607,216) and a portion of Guanine Nucleotide Binding Protein (G Protein) α 12, GNA12 gene (* 604,394) (Table 1). CAspase Recruitment Domain family member 11 (CARD11) encodes a protein that belongs to the membrane-associated guanylate kinase (MAGUK) family. This class of proteins functions as a scaffold for nuclear factor kappa-B involved in the regulation of peripheral B-cell differentiation and in different critical T-cell effector functions. Allelic variants in this gene have been associated to immune system disorders (# 616,452—immunodeficiency 11a; # 615,206—B-cell expansion with Nfkb and T-cell anergy; # 617,638—immunodeficiency 11b with atopic dermatitis).

GNA12 encodes a Guanine nucleotide-binding protein (G protein). The G proteins are involved as modulators or transducers in various transmembrane signalling systems. A role of this gene has been hypothesized in spermatogenesis [17] and in preeclampsia [18]. In contrast, association with neurodevelopment disorders or congenital malformations have not been reported.

The SDK1 gene encodes for a homophylic adhesion molecule, a transmembrane protein localized in synaptic membranes. SDK1 is highly concentred at synapsis where it binds scaffolding proteins of the MAGI family, increasing their concentration at synaptic sites [19]. The extracellular domains of SDK protein mediate the homophilic adhesion, promoting the formation of synapsis only between SDK-positive cells, then supporting the laminar-specific process and the synaptic specificity [20]. Several genome-wide association studies in humans identified SDK1 gene as a risk factor for different neurodevelopmental and psychiatric disorders, such as ASD [21,22,23,24,25,26], an attention-deficit hyperactivity disorder [27] and schizophrenia [28]. In a recent paper, Corley et al. analysed the DNA methylation status in the subventricular zone from post-mortem human brains in an autistic cohort and in a healthy control group [29]. They found that in only a few genes CpG methylation levels were always consistently altered in the brains of ASD patients independently of their age. Among these genes, a significant reduction in the methylation levels was detected at several CpG sites of *SDK1* gene. This reduced methylation is probably associated with an increased expression of such gene. Interestingly, the duplication detected in our two patients contains three CpG Islands located in the SDK1 promoter. We also analysed duplications carried by patients reported in Table 1 and observed that CpG islands also occur in all duplications involving SDK1 and carried by patients with ASD and/or a neurodevelopmental delay (Table 1 and Figure 1).

In a recent work carried out by Shi et al., in order to better characterize mechanisms by which CNVs affect cellular phenotype, the authors tested whether inter-individual differences in DNA copy number were associated with inter-individual variation in DNA methylation levels detected across the human genome. Using paired CNV and methylation data from the 1000 genomes and HapMap projects, the authors found that several CNVs were associated with methylation of multiple CpG sites and vice versa. CNV-associated methylation changes were correlated with gene expression. Some of these CNVs were (i) enriched for regulatory regions, transcription factor-binding sites; (ii) involved in long-range physical interactions; (iii) associated with methylation of imprinted gene and (iv) among those previously reported by several genome-wide association studies (GWASs). These results clearly suggest that structural variations together with methylation may affect cellular phenotype [30]. Then, we can assume that duplications of SDK1 regions overlapping with the CpG island can interfere with the correct methylation status of SDK1 and contribute to the onset of a neurodevelopmental phenotype. Further functional studies have thus to be carried out to better clarify the effect of the 7p22.2 microduplications detected in our patients and to provide insight into the genetic basis of the 7p22 duplication syndrome.

## 5. Conclusions

In this study, we describe two patients carrying a microduplication including only the 7p22.2 sub-band. To date, only Cox and Butler (2015), suggested a role of this sub-band in the 7p22 microduplication syndrome. Our data support this hypothesis. Further, we suggest that the rearrangement of this sub-band involving SDK1 gene can be associated with a neurodevelopmental phenotype, mainly characterized by speech delay, psychomotor delay, and autistic traits, with a high frequency of mild dysmorphic traits. SDK1 is a gene implicated in the epigenetic delay in autistic subjects [29]. Therefore, we speculate that the detected structural variations together with methylation may affect SDK1 expression which could significantly impact disease pathogenesis.

## Figures and Tables

**Figure 1 genes-14-01292-f001:**
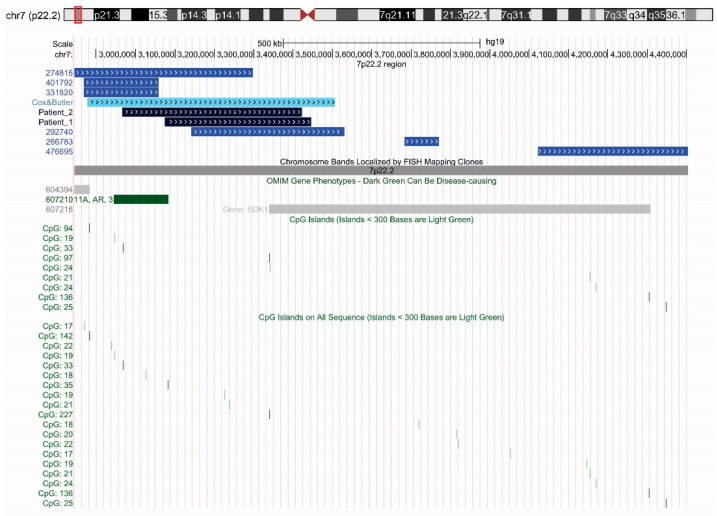
Schematic representation of the microduplications restricted to the only 7p22.2 sub-band gathered from DECIPHER (medium blue), Cox and Butler [10] (light blue), and in this study (dark blue).

**Table 1 genes-14-01292-t001:** Clinical features of patients carrying microduplications restricted to the only 7p22.2 sub-band gathered from DECIPHER, Cox and Butler [10] and in this study.

Cases	Source	Position	Size	Phenotype	Genes	CNV Classification (Decipher)	SDK1 Region Involved
Patient 1	Present study	3,074,785–3,445,950	371 kb	Severe expressive and receptive language delay, ASD dysmorphic traits.	CARD11, SDK1		Exon 1 and intron 1
Patient 2	Present study	2,967,525–3,422,668	371 kb	Psychomotor delay and severe speech delay; minor skeletal abnormalities; mild dysmorphic traits.	CARD11, SDK1		Exon 1 and intron 1
Cox and Butler	Cox and Butler [10]	2,878,677–3,507,572	629 kb	Developmental and speech delay. Dysmorphic traits.	GNA12, CARD11, SDK1		Exon 1 and intron 1
292740	DECIPHER	3,142,074–3,531,219	389 kb	Cognitive impairment.	SDK1	Uncertain	Exon 1 and intron 1
266783	DECIPHER	3,684,139–3,771,337	87 kb	ASD; intellectual disability.	SDK1	Uncertain	IOntron 4
476695	DECIPHER	4,022,953–4,508,102	485 kb	No phenotypes have been entered.	SDK1	Uncertain	From exon 14 to end of the gene
274816	DECIPHER	2,845,372–3,298,538	453 kb	Generalized tonic seizure.	GNA12, CARD	Not reported	
401792	DECIPHER	2,868,939–3,058,828	189.9 kb	Abnormality of the musculoskeletal system, of the eye, of the nervous system and facial dysmorphism.	GNA12, CARD11	Uncertain	
331820	DECIPHER	2,868,939–3,058,823	189.88 kb	Strabismus, wide nasal base, scoliosis, peripheral demyelination.	GNA12,CARD11	Not reported	

## Data Availability

The data that support the findings of this study are available on request from the corresponding author.

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
