# Peer review of "7p22.2 Microduplication: A Pathogenic CNV?"

_genes, 2023, doi:10.3390/genes14061292_

Round 1

Reviewer 1 Report

The paper is interesting and brings new information.

The text is pleasant and leads the reader along the line of thought presented.

-proofread English.
- improve table 1. Patients from the present study and from Cox and Butler are described, but it is not clear whether the remaining ones are also from Cow and Butler.

Author Response

The paper is interesting and brings new information.The text is pleasant and leads the reader along the line of thought presented.

We would thank the Reviewer for appreciating our work.

-proofread English.

The Ms. has been revised by a mother tongue English speaker in order to check  and correct  grammatical errors.

- improve table 1. Patients from the present study and from Cox and Butler are described, but it is not clear whether the remaining ones are also from Cow and Butler.

We thank the referee for this suggestion. In the revised version of our Ms. the Table 1 now also includes this information.

Reviewer 2 Report

The authors present potentially interesting cases that could shed some more light on 7p22.2 microduplication as a pathogenic CNV. I think the description of these cases is a relevant contribution to the scientific literature and the report is clearly written. There are, however, some issues that need to be addressed.

The authors rely heavily on the paper by Cox and Butler from 2015, but the phenotype of the patient described here seems quite different from the patients described in this report. I feel this should be emphasized more and discussed at some more length in the discussion. Also, a clearer conclusion about the different genes involved in the different patients (case reports, database and Cox and butler) and the associated phenotypes would benefit the report. Moreover, I feel that analysis of public databases is parts of the results and should be presented there.

It is not completely clear to me how increased copies of the SDK1 CpG islands would lead to a change in the SDK1 methylation status. The authors should elaborate and give some references to literature on how this works. If this is not clearly established I would rephrase the conclusion to something a bit less definitive.

-

Author Response

The authors rely heavily on the paper by Cox and Butler from 2015, but the phenotype of the patient described here seems quite different from the patients described in this report. I feel this should be emphasized more and discussed at some more length in the discussion. Also, a clearer conclusion about the different genes involved in the different patients (case reports, database and Cox and butler) and the associated phenotypes would benefit the report. Moreover, I feel that analysis of public databases is parts of the results and should be presented there.

We thank the referee for all these suggestions. The comparison between the clinical phenotype of the patients described in our and in other works has been described adding further details and emphasizing similarities and differences [see lines 170-173 and 184-188].

We revised the “Conclusion” section also in the light of the results of the study focused on the characterization of mechanisms by which CNVs affect cellular phenotype (Shi et al., Genomic Medicine, 2020) [see lines 248-249].

Table 1 has been moved in the “Results” section on the revised manuscript.

It is not completely clear to me how increased copies of the SDK1 CpG islands would lead to a change in the SDK1 methylation status. The authors should elaborate and give some references to literature on how this works. If this is not clearly established I would rephrase the conclusion to something a bit less definitive.

We revised the “Discussion” and the “Conclusion” sections also in the light of the results of the study focused on the characterization of mechanisms by which CNVs could affect the observed clinical phenotype on the analysed patients (Shi et al., Genomic Medicine, 2020). In fact, it has been demonstrated that using paired CNV and methylation data from the 1000 genomes and HapMap projects, several CNVs were associated with methylation of multiple CpGs and vice versa. These CNVs were (i) correlated with gene expression; (ii) enriched for regulatory regions, transcription factor-binding sites, (iii) involved in long-range physical interactions; (iv) associated with methylation of imprinted gene and (v) among those previously reported by genome-wide association studies [see lines 224-235 and 248-249].